# Identification of Pre-Diabetic Biomarkers in the Progression of Diabetes Mellitus

**DOI:** 10.3390/biomedicines10010072

**Published:** 2021-12-30

**Authors:** Jae-Ho Lee, Do-Young Kim, Rubee Pantha, Eun-Ho Lee, Jae-Hoon Bae, Eugene Han, Dae-Kyu Song, Taeg Kyu Kwon, Seung-Soon Im

**Affiliations:** 1Department of Physiology, Keimyung University School of Medicine, Daegu 42601, Korea; hoyaya86@naver.com (J.-H.L.); kkdy0512@gmail.com (D.-Y.K.); rubs.angel46@gmail.com (R.P.); yeh0322@naver.com (E.-H.L.); jhbae@dsmc.or.kr (J.-H.B.); dksong@dsmc.or.kr (D.-K.S.); 2Department of Internal Medicine, Division of Endocrinology, Keimyung University School of Medicine, Daegu 42601, Korea; eghan@dsmc.or.kr; 3Department of Immunology, Keimyung University School of Medicine, Daegu 42601, Korea; kwontk@dsmc.or.kr

**Keywords:** diabetes mellitus, type 2 diabetes mellitus, pre-diabetes, biomarkers, metabolic disease

## Abstract

Type 2 diabetes mellitus (T2DM) is a major global health issue. The development of T2DM is gradual and preceded by the pre-diabetes mellitus (pre-DM) stage, which often remains undiagnosed. This study aimed to identify novel pre-DM biomarkers in a high-fat diet (HFD)-induced pre-DM mouse model. Male C57BL/6J mice were fed either a chow diet or HFD for 12 weeks. Serum and liver samples were isolated in a time-dependent manner. Semi-quantitative assessment of secretory cytokines was performed by cytokine array analysis, and 13 cytokines were selected for further analysis based on the changes in expression levels in the pre-DM and T2DM stages. HFD-fed mice gained body weight and exhibited high serum lipid, liver enzyme, glucose, and insulin levels during the progression of pre-DM to T2DM. The mRNA expression of inflammatory and lipogenic genes was elevated in HFD-fed mice The mRNA expression of Fc receptor, IgG, low affinity Iib, lectin, galactose binding, soluble 1, vascular cell adhesion molecule 1, insulin-like growth factor binding protein 5, and growth arrest specific 6 was elevated in the pre-DM, which was confirmed by measuring protein levels. Our study identified novel pre-DM biomarkers that may help to delay or prevent the progression of T2DM.

## 1. Introduction

Diabetes mellitus (DM) is a highly prevalent metabolic disease characterized by uncontrolled elevation of blood glucose levels, which leads to multiple complications, morbidity, and mortality [1,2,3]. DM is predominantly categorized as type 1 diabetes mellitus (T1DM) or type 2 diabetes mellitus (T2DM) [2,3]. T2DM is the most common type of DM, and its global prevalence is steadily increasing [3,4]. Pre-diabetes mellitus (pre-DM) is a metabolic state that is closely associated with T2DM, in which plasma glucose levels lie above the range of normoglycemia but below diabetes thresholds [5,6,7]. Indeed, pre-DM has been termed “Intermediate hyperglycemia” by the World Health Organization and “High Risk State of Developing Diabetes” by the International Expert Committee congregated by the American Diabetes Association [8]. Globally, the number of patients with pre-DM is increasing rapidly, and the prevalence of pre-DM is expected to increase to 470 million or higher by 2030 [8]. Individuals with pre-DM have a high risk of developing T2DM [6,8,9]. In total, 5–10% of patients who progress to DM per year present with pre-DM-related complications [6,7]. In pre-DM, abnormalities such as insulin resistance (IR) and β-cell dysfunction persist simultaneously prior to detectable changes in glucose levels [8]. Observational evidence supports an association of pre-DM with early stages of nephropathy, diabetic retinopathy, small fiber neuropathy, and the risk of macrovascular disease [8,9].

Obesity is a prime risk factor for the progression of pre-DM to T2DM [10,11]. Epidemiological studies have indicated that an increase in dietary fat consumption promotes obesity [12]. However, obesity alone is insufficient to explain disease progression, as only a small proportion of obese individuals have T2DM. In this regard, genetic factors likely contribute to the progression of pre-DM to T2DM. Relevant animal models of various progressive stages of T2DM development (i.e., from insulin-resistant pre-DM to T2DM) may facilitate the elucidation of associated genetic signatures in insulin-sensitive tissues and reveal the molecular mechanisms underlying disease progression [13].

To induce pre-DM, various animal models, such as high-fat/high-calorie diet-fed rodents and high-fat diet (HFD)-fed streptozotocin (STZ)-injected canine models, have been established [14]. The progressive stages of T2DM in HFD models are classified as 6 weeks (pre-DM), 12 weeks (T2DM), and 20 weeks (late-stage T2DM) [13]. As the progression of pre-DM results in T2DM, detailed investigations of pre-DM and associated biomarkers may attenuate the risk of progression to T2DM.

Biomarkers generally represent features that indicate pathogenic processes, normal biological processes, and/or pharmacological reactions to clinical treatment. HbA1c is a well-established, traditional biomarker. MicroRNAs and several proteomic and cytokine markers are emerging biomarkers, referred to as novel biomarkers [3].

Given that the development of T2DM is preceded by pre-DM, which is often undiagnosed [15], investigations of pre-DM and associated biomarkers may help to delay or prevent progression to T2DM. Therefore, the aim of this study was to identify novel pre-DM biomarkers using an HFD-induced pre-DM mouse model. This approach revealed various putative pre-DM and T2DM biomarkers that may facilitate the diagnosis of pre-DM and delay or prevent the development of T2DM.

## 2. Materials and Methods

### 2.1. Animal Studies

Eight-week-old C57BL/6J male mice were housed in specific-pathogen-free mouse facilities under 12:12 h light–dark cycles (6 a.m.–6 p.m. light, 6 p.m.–6 a.m. dark) at 23 °C with 60–70% humidity and free access to water. The liver sample sizes of the animal experiments were estimated according to the known variability of the assays. These mice were fed either a chow diet as control or an HFD (D12492, containing 60% fat (kcal%); Research Diet, New Brunswick, NJ, USA) for 12 weeks. Body weights of control and HFD-fed mice were monitored at 1, 2, 4, 6, 8, and 12 weeks. Body composition was determined using nuclear magnetic resonance imaging (LF50 BCA-Analyzer, Bruker, Brussels, Belgium). Fat mass and lean mass were analyzed by dividing the fat or lean weight of individual mice by body weight. Mice were anesthetized with isoflurane (Hana Pharm. Co., Gyeonggi-DoHwaseong-si, Korea) and sacrificed at 9:00 a.m. during the light cycle. All animal experiments were performed at Keimyung University School of Medicine, Daegu, South Korea, following the protocol approved by the Institutional Animal Care and Use Committee of Keimyung University (KM-2016-22R1).

### 2.2. Serum Profiling

Serum was collected from the control and HFD-induced pre-DM mice at 1, 2, 4, 6, 8, and 12 weeks to measure the levels of various serum parameters during the progression of pre-DM to T2DM. Initially, serum triglycerides (TGs) were extracted using McGowan’s method as described previously [16]. Serum TG levels were quantified using a TG assay kit (AM 157S-K, Asan Pharm Co., Seoul, Korea), and the absorbance of the samples was measured at 500 nm using Infinite^®^ 200 PRO (Tecan Trading AG, Männedorf, Switzerland). Similarly, cholesterol was extracted using Richmond’s method, as described previously [17]. Serum cholesterol levels were quantified using a total cholesterol assay kit (AM 202-K, Asan Pharm Co.), and the absorbance of samples and standard was measured at 550 nm. Hepatic injury was determined by measuring levels of serum alanine aminotransferase (ALT) and aspartate aminotransferase (AST) using spectrophotometric assay kits (AM 101-K, Asan Pharm Co., Seoul, Korea). The absorbance values of serum biochemicals were measured at a wavelength of 490 nm. Serum glucose levels were measured using the Invitrogen Glucose Colorimetric Detection Kit (EIAGLUC, Carlsbad, CA, USA). Plasma insulin levels were analyzed using a commercially available ELISA kit (80-INSMS-E01, Alpco Diagnostics, Salem, NH, USA), following the manufacturer’s instructions.

### 2.3. Analysis of the Cytokine Secretome

To evaluate and compare the diversity of the secretory cytokine profiles, serum semi-quantitative assessment of 97 cytokines was performed in the control and HFD-induced pre-DM mice. The levels of cytokines in the serum were measured using a mouse cytokine antibody array kit (RayBio Mouse Cytokine Antibody Array G series II, RayBiotech, Inc., Norcross, GA, USA). In addition, cytokine profiling antibody arrays were performed using a customized service provided by eBiogen Inc. (Seoul, Korea).

### 2.4. Quantitative Polymerase Chain Reaction (qPCR)

Total RNA was obtained from liver tissues of control and HFD-fed mice using TriZol (Invitrogen), and cDNA was synthesized using the iScript™ cDNA synthesis kit (Bio-Rad Laboratories, Hercules, CA, USA). qPCR was performed using the CFX96^TM^ real-time PCR system (Bio-Rad Laboratories) to measure the mRNA expression levels of various genes. The mRNA levels were normalized to the expression of ribosomal protein L32 by calculating the delta–delta threshold cycle method. The primer sequences for the genes used in qPCR are listed in Table 1.

### 2.5. Immunoblotting

Protein samples were extracted from the liver tissues of control and HFD-fed mice. Briefly, liver tissue lysates were extracted using T-PER™ (Tissue Protein Extraction Reagent; Thermo Scientific, Rockford, IL, USA) supplemented with a protease and phosphatase inhibitor (Thermo Scientific). Proteins were diluted with 5× sample buffer (EBA-1052, ELPIS BIOTECH, Seoul, Korea) and heated at 95 °C for 5 min. The proteins were separated by 5–15% Tris-HCl SDS/PAGE gel electrophoresis and transferred onto nitrocellulose membranes (GE Healthcare, Uppsala, Sweden). All immunoblots were developed using HRP-conjugated secondary antibody with enhanced chemiluminescence (Clarity™ Western ECL Substrate, Bio-Rad Laboratories). Subsequently, protein bands were detected using a chemiluminescence imaging system (Fusion Fx, Vilber Lourmat, Eberhardzell, Germany). Fatty acid synthase (FAS, 3180), acetyl-CoA carboxylase 1 (ACC1, 4190), stearoyl-CoA desaturase-1 (SCD1, 2794), and lectin, galactose binding, and soluble 1 (Galectin-1/LGALS1, 5418) antibodies were purchased from Cell Signaling Technology, Inc. (Beverly, MA, USA). Vascular cell adhesion molecule 1 (VCAM1, ab115135) was purchased from Abcam (Cambridge, 0AX, UK). Fc receptor, IgG, low affinity Iib (FCGR2B, MBS7045652), and growth arrest-specific 6 (GAS6, MBS9129771) were purchased from MyBioSource (San Diego, CA, USA). Insulin-like growth factor binding protein 5 (IGFBP5, VB2925623) was purchased from Invitrogen. All protein bands were normalized to β-actin (A5441; Sigma-Aldrich, St. Louis, MO, USA).

### 2.6. Statistical Analysis

All data were analyzed using GraphPad Prism 9.1 software (GraphPad Software Inc., San Diego, CA, USA) and expressed as mean ± SD of the number of determinations carried out in triplicate. Variables were tested for normality and then the different groups were compared using the paired sample *t*-test, where *p* < 0.05 was considered statistically significant between groups.

## 3. Results

### 3.1. Monitoring of Body Weight, Fat Mass, and Lean Body Mass after 12-Week High-Fat Diet

To investigate pre-DM and T2DM-associated changes, C57BL/6J mice were fed either a chow diet or HFD for 12 weeks to induce pre-DM and T2DM, respectively. Body weight, fat mass, and lean mass were measured at 1, 2, 4, 6, 8, and 12 weeks in both control and HFD-fed mice. Compared to that of control mice, weight of the HFD-fed mice increased significantly after 1 week, which continued thereafter until 12 weeks (Figure 1A). The increase in body weight in HFD-fed mice was because of an increase in fat mass and a decrease in lean mass (Figure 1B,C).

### 3.2. DM-Associated Serum Parameters in Pre-DM

To examine the serum levels of lipids, liver enzymes, glucose, and insulin in our control and HFD-induced pre-DM mouse model, blood was collected at 1, 2, 4, 6, 8, and 12 weeks. Pre-DM and T2DM are often associated with lipid abnormalities [18,19]. TG and cholestrol levels were significantly higher in HFD-fed mice after 4 weeks than in control mice at the respective weeks (Figure 2A,B). Serum levels of the liver enzymes alanine aminotransferase (ALT) and aspartate aminotransferase (AST) were significantly higher in HFD-fed mice after 6 and 8 weeks, respectively, than in control mice at the respective weeks, implying early signs of hepatocellular injury (Figure 2C,D). Similarly, serum glucose and insulin levels were significantly higher in HFD-induced pre-DM mice at 8 weeks than in control mice at the respective weeks, indicating pre-DM and preliminary stages of IR (Figure 2E,F).

### 3.3. Secretome Analysis of Serum in HFD-Induced Pre-DM and DM

To assess the expression of various secretory cytokines in pre-DM, cytokine array analysis was performed on serum samples from control and HFD-induced pre-DM mice. Of the 97 cytokines analyzed in this study, 13 were selected for analysis based on changes in expression levels in pre-DM and T2DM stages. Table 2 presents a list of selected cytokines and their expression in the secretome. Differentially expressed cytokines in serum are presented in a heatmap (Figure 3A). The selected cytokines in pre-DM and T2DM stages are depicted in a scatter plot, with each dot representing a cytokine (Figure 3B). The number of differentially expressed cytokines overlapping at 6, 8, and 12 weeks following HFD is shown in a Venn diagram (Figure 3C). Protein–protein interaction between the selected cytokines was analyzed using STRING (Figure 3D). Our analyses revealed that secretion of the selected cytokines increased during the progression from pre-DM to T2DM.

### 3.4. Lipogenic Gene Expression in the Liver of HFD-Induced Pre-DM and DM Mice

Serum lipid levels increased in HFD-induced pre-DM mice. To investigate the expression levels of lipogenic genes in the pre-DM stage, mRNA and protein levels of lipogenic genes were measured in the liver samples of control and HFD-induced pre-DM mice. mRNA expression of sterol-regulatory-element-binding protein-1c (Srebp-1c), a transcription factor involved in the regulation of genes responsible for lipid metabolism [20], was significantly upregulated in HFD-induced pre-DM mice after 2 weeks compared with that in control mice at the respective weeks (Figure 4A). Expression of the target genes of Srebp-1c, such as Fas, Acc1, and Scd1, was measured. mRNA expression of Fas and Scd1 was significantly higher in HFD-induced pre-DM mice at 2 weeks and after 6 and 8 weeks, respectively, compared with that in control mice at the respective weeks (Figure 4B,D). Similarly, the mRNA expression of Acc1 was significantly increased in HFD-induced pre-DM mice after 8 weeks compared with that in control mice at the respective weeks (Figure 4). Moreover, protein levels of these lipogenic genes, including FAS, ACC1, and SCD-1, were also increased in pre-DM and T2DM stages in HFD-fed mice (Figure 4E).

### 3.5. Inflammatory Gene Expression in Pre-DM and DM

Pre-DM and IR are characterized by distinct inflammatory states [5]. Hepatic inflammation can also lead to IR [21]. Therefore, to elucidate the inflammatory state in pre-DM, mRNA expression of inflammatory and anti-inflammatory genes was measured in control and HFD-induced pre-DM mice. mRNA expression of pro-inflammatory genes such as monocyte chemoattractant protein 1 (Mcp1) was significantly upregulated (Figure 5A), whereas mRNA expression of tumor necrosis factor alpha (Tnf-α) and interleukin-1β (Il-1β) was upregulated after 2 weeks in HFD-induced pre-DM mice compared to that in control mice at the respective weeks (Figure 5B,C). mRNA expression of anti-inflammatory genes was also measured in control and HFD-induced pre-DM mice. mRNA expression of anti-inflammatory genes, such as mannose receptor c-type 1 (CD206), was downregulated in HFD-induced pre-DM mice after 2 weeks, compared with that in control mice at the respective weeks (Figure 5D), whereas inflammatory zone 1 (Fizz1) and c-type lectin domain containing 10, member A (Clec10a) was downregulated after 4 weeks, compared with that in control mice at the respective weeks (Figure 5E,F).

### 3.6. Characterization of Pre-DM Biomarkers in the Liver

To identify pre-DM biomarkers, mRNA and protein levels of selected genes were measured in the liver samples of control and HFD-fed mice. mRNA expression of several genes was increased in the pre-DM stage; these genes were identified as novel pre-DM biomarkers. mRNA expression of Lgals1 and Fcgr2b was significantly upregulated in HFD-induced pre-DM mice after 6 and 8 weeks, respectively, compared with that in control mice at the respective weeks (Figure 6A,B). Similarly, mRNA expression of Vcam1 was significantly higher in HFD-induced pre-DM mice from the first week to the progression from pre-DM to T2DM stage compared with that in control mice at the respective weeks (Figure 6C). Similarly, mRNA expression of Igfbp5 and Gas6 was significantly upregulated in HFD-induced pre-DM mice in the first week and after 6 weeks compared with that in control mice at the respective weeks (Figure 6D,E). Protein levels of FCGR2b, LGALS1, VCAM1, IGFBP5, and GAS6 increased after 6 weeks in HFD-induced pre-DM mice (Figure 6F). Furthermore, mRNA expression of pre-DM marker genes was measured in the liver samples from ob/ob and db/db mice. In both ob/ob and db/db mice, mRNA expression of Lgals1, Vcam1, and Igfbp5 was significantly increased, whereas the mRNA expression of Fcgr2b was significantly decreased. mRNA expression of Gas6 was significantly increased in ob/ob mice, but was not altered in db/db mice (Appendix A).

### 3.7. Identification of DM Marker Genes in the Liver at T2DM Stage

To confirm and identify DM marker genes, mRNA levels of selected genes were measured in liver samples from control and HFD-induced pre-DM mice. mRNA expression of several genes was increased at the T2DM stage in HFD-induced pre-DM mice; these genes were identified as putative biomarkers of T2DM. In the HFD-induced pre-DM mouse model, mRNA expression of cytokines, such as interleukin-12b (Il-12b) and interleukin-21 (Il-21), was significantly increased after 8 weeks compared to that in control mice after 8 weeks (Figure 7A,E). Similarly, mRNA expression of the TNF receptor superfamily-coding gene, Tnfrsf1b, was significantly increased after 8 weeks (Figure 7B), whereas Tnfrsf12a and Tnfrsf18 mRNA levels were elevated at 12 weeks in the HFD-induced pre-DM mouse model (Figure 7C,D). mRNA expression of AXL receptor tyrosine kinase (Axl) and insulin like growth factor-6 (Igfbp6) was significantly elevated after 8 weeks (Figure 7F,G). mRNA expression of chemokine (C-X-C motif) ligand 11 (Cxcl11) was significantly upregulated after 6 weeks in the HFD-induced pre-DM mouse model (Figure 7H). Moreover, mRNA expression of DM marker genes was measured in liver samples collected from ob/ob and db/db mice (Appendix A).

## 4. Discussion

T2DM is a serious chronic metabolic disease that has become a major global health issue. T2DM develops gradually and is preceded by the pre-DM stage, which is often undiagnosed [15,22]. T2DM and pre-DM are components of metabolic disorders, which significantly overlap [6]. However, studies on pre-DM and the associated biomarkers are limited. In this study, we identified major pre-DM and T2DM biomarkers in an HFD-induced pre-DM mouse model compared to standard chow diet-fed control mice. Mice fed an HFD are well established as an experimentally induced pre-DM animal model [14]. The progression of T2DM occurs in stages of 6 weeks (pre-DM stage), 12 weeks (T2DM stage), and 20 weeks (late-stage T2DM) [13]. Obesity is a major contributing factor to the progression from pre-DM to T2DM [10,11]. In this regard, HFD-fed mice exhibited significant body weight gain compared to control mice, which may have been due to an increase in fat mass and concomitant decrease in lean mass.

β-cell dysfunction has been associated with increased serum TG levels in pre-DM [5]. In this study, serum levels of lipids, such as TG and cholesterol, increased from the pre-DM to T2DM stages. Previous studies have highlighted an association of T2DM with increased TG levels [23]. Increased TG levels indicate pre-DM and DM disease, as it is an indirect sign of insulin resistance, which is also responsible for NAFLD development and its progression to non-alcoholic steatohepatitis (NASH) [24]. Moreover, increased lipogenic gene expression may promote TG synthesis [25]. HFD-induced pre-DM mice exhibited significantly elevated mRNA and protein levels of lipogenic genes in the liver compared to control mice. Moreover, serum glucose and insulin levels were increased in pre-DM mice. Both hyperglycemia and hyperinsulinemia aggravate IR and stimulate hepatic lipogenesis [26]. A recent study demonstrated that an increase in serum insulin and glucose levels may underpin elevated serum TG and cholesterol levels [27]. Collectively, these data demonstrate that an increase in serum glucose and insulin levels elevate lipogenic gene expression and serum lipid levels, indicating an increased risk of developing T2DM.

IR in the liver is the predominant contributor to metabolic syndrome, which includes disorders such as T2DM [26]. IR and T2DM are associated with elevated ALT and AST enzyme levels [28]. Indeed, serum levels of the liver enzymes, ALT and AST, were significantly increased in HFD-induced pre-DM mice, implying early signs of hepatic dysfunction. A human study reported that increased levels of liver enzymes are associated with an increased risk of T2DM [28]. Moreover, T2DM is characterized by low-grade inflammation [29]. Increased serum transaminase levels are also explained by the low-grade inflammation present in DM, which is one of the main pathophysiological mechanisms underlying NASH development [30]. Low-grade inflammation, caused by pro-inflammatory cytokines, is also implicated in the development of endothelial dysfunction, which can occur in pre-DM patients, thus highlighting the importance of detecting pre-DM as quickly as possible, as complications begin to develop in this phase [31]. Inflammation plays a key role in the development of IR in individuals with obesity. In this study, the expression of pro-inflammatory genes, such as *Mcp1*, *Tnfα*, and *Il-1β*, was increased in HFD-fed mice. MCP-1 links obesity to IR, and plasma levels of MCP-1 are increased in T2DM [32]. TNFα, which is associated with obesity, has been linked to the onset of IR and T2DM [28,33]. Previous studies have reported increased levels of TNFα in serum samples of individuals with pre-DM [34]. Additionally, IL-1β is a macrophage-induced immune system modulator that decreases βcell activity and insulin sensitivity [35]. Overall, these findings suggest that elevated levels of liver enzymes and inflammation lead to the progression of T2DM.

To identify pre-DM and T2DM biomarker genes, semi-quantitative assessment of 97 cytokines in serum samples from control and HFD-induced pre-DM mice was performed. Of these cytokines, 13 were selected for subsequent analysis of mRNA expression in the liver samples. mRNA expression of several genes was elevated in the pre-DM and T2DM stages, respectively. In this study, HFD-fed mice showed significantly increased mRNA and protein levels of Fcgr2b, Lgals1, Vcam1, Igfbp5, and Gas6 at the pre-DM stage compared to those in control mice. FCGR2B is a member of the immune receptor IgG Fc gamma receptor family. A recent study reported the role of the FcgR family in the development of T2DM [36]. Similarly, increased levels of LGALS1 in the subcutaneous adipose tissue of individuals with T2DM have been observed [37]. Studies have reported elevated Vcam1 and Igfbp5 expression in association with high glucose levels [38,39]. GAS6 is a vitamin K-dependent cytokine for receptor tyrosine kinases such as AXL. However, the association among GAS6, IR, and T2DM remains controversial [40]. The elevated mRNA and protein levels of these genes at the pre-DM stage in HFD-fed mice compared to those in control mice highlight their potential utility as novel pre-DM biomarkers.

Genes with elevated mRNA expression at the T2DM stage in HFD-induced pre-DM mice may act as novel T2DM biomarkers. The expression of known biomarkers of T2DM (i.e., cytokine *Il-12b*) was significantly upregulated after 8 weeks. A recent study on patients with chronic plaque psoriasis revealed that variation of *IL-12b* influences T2DM [41]. Moreover, Il-12, consisting of an Il-12b allele, has been previously shown to promote the development of T1DM in nonobese diabetic mice [42,43]. Several studies have reported an association of Il-12b with T1DM [40]. Similarly, HFD-induced pre-DM mice exhibited increased mRNA expression of *Tnfrsf1b* and *Tnfrsf12a* at the T2DM stage. Previous studies have demonstrated an association of TNFRSF1B genotype with clinical neuropathy in T2DM [44]. However, Tabassum et al. reported a lack of association between TNFRSF1B and T2DM in patients from North India [29,45]. TNFRSF12A (also known as fibroblast growth factor-inducible 14) has been associated with an increased risk of T2DM in women [46]. Furthermore, the expression of TNFRSF18, a glucocorticoid-induced tumor necrosis factor receptor (GITR), is upregulated during inflammation. In this study, we observed increased mRNA expression of *Tnfrsf18* at the T2DM stage. However, this result is in contrast with previous findings, suggesting a protective role of GITR in the development of T2DM [47].

mRNA expression of *Il-21* was upregulated after 8 weeks in HFD-fed mice. Studies on IL-21 receptor-deficient mice have reported complete termination of T1DM development [48]. Furthermore, mRNA expression of *Axl* was increased at the T2DM stage in HFD-induced pre-DM mice. These findings agree with previous reports of increased expression of Axl and Gas6 in STZ-induced diabetic rats [49]. IGFBPs are associated with IR and diabetic complications [50]. In this study, increased expression of *Igfbp6* was observed after 8 weeks in HFD-fed mice. However, a study by Bergman et al. on STZ-induced diabetic rats reported downregulated Igfbp6 expression in the eye [51]. In contrast to the findings of Bergman et al., Lu et al. reported that serum levels of IGFBP6 were increased in T1DM patients [50]. Although IL-21, AXL, and IGFBP6 are strongly implicated in the progression of DM, increased expression of Il-21, Axl, and Igfbp6 in T2DM has not been reported to date.

The chemokine (C-X-C motif) ligand (CXCL) family plays a vital role in inflammation [52]. mRNA expression of *Cxcl11* was increased in the T2DM stage in HFD-induced pre-DM mice. A recent study reported a close association between CXCL11 and inflammation in the progression of IR in obese individuals [53]. Collectively, this model exhibited increased mRNA expression of *Il-12b*, *Tnfrsf1b*, *Tnfrsf12a*, *Tnfrsf18*, *Il-21*, *Axl*, *Igfbp6*, and *Cxcl11* at the T2DM stage, suggesting their potential role as T2DM biomarkers.

In conclusion, this study identified novel pre-DM and T2DM biomarkers in a mouse model of HFD-induced pre-DM. Our findings provide support for the association between obesity and various serum parameters in the progression of pre-DM to T2DM. Of note, we identified increased expression of lipogenic genes and inflammation in our pre-DM mouse model. Nevertheless, prospective studies are warranted to confirm these findings and elucidate the roles of these genes in the risk of developing pre-DM and T2DM. Overall, our findings may facilitate the diagnosis of pre-DM and help to delay or prevent the progression of T2DM.

## Figures and Tables

**Figure 1 biomedicines-10-00072-f001:**
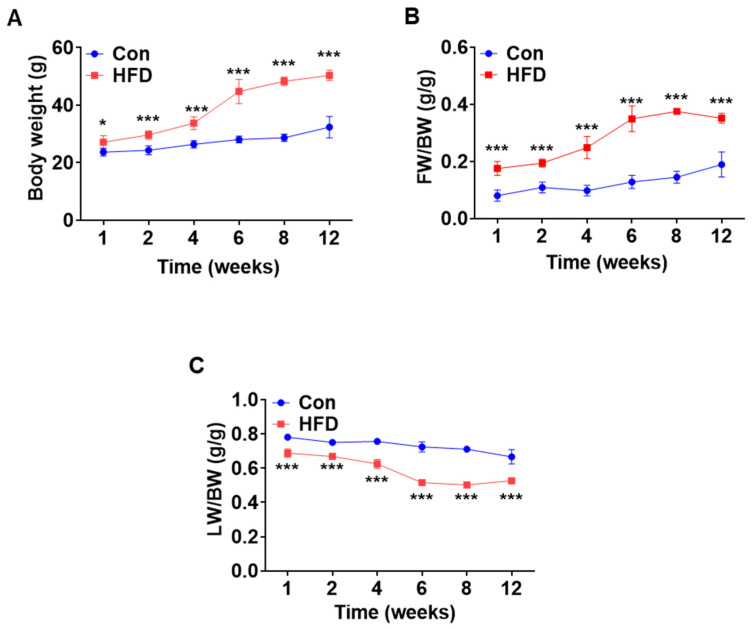
High-fat diet induced pre-diabetes and diabetes mellitus mouse model. Normal mice were fed either chow diet or HFD for 12 weeks (*n* = 5). (**A**) Changes in body weight over a period of 12 weeks. (**B**,**C**) Time course of changes in fat mass and lean mass. * *p* < 0.05 and *** *p* < 0.001 Control vs. HFD-fed mice at the respective weeks. HFD: high-fat diet.

**Figure 2 biomedicines-10-00072-f002:**
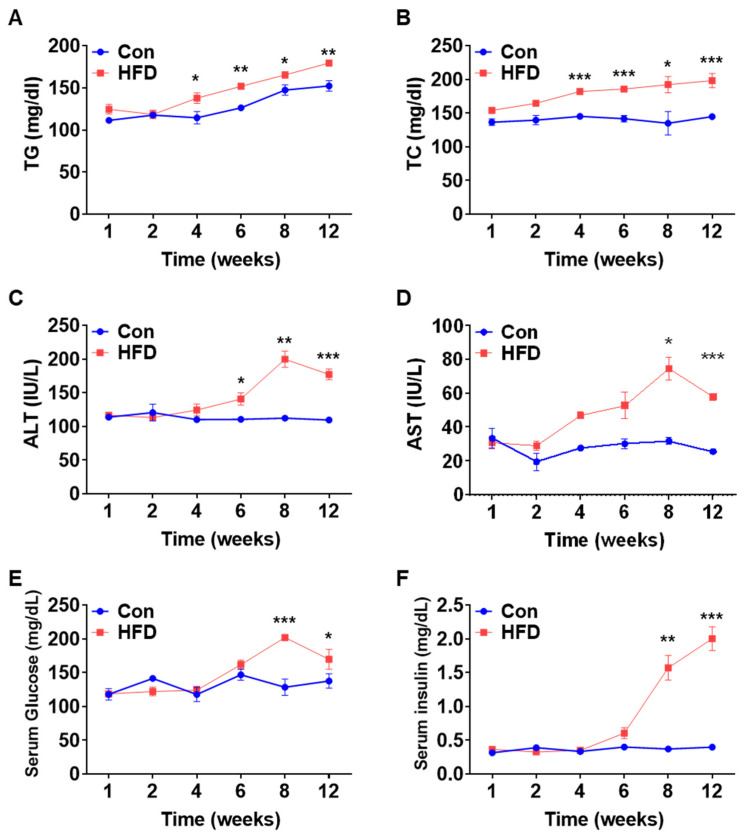
Changes in serum profile. Serum lipids and liver toxicity marker profiling analysis of WT mice after feeding chow diet or HFD for 12 weeks: (**A**) TG, (**B**) TC, (**C**) ALT, (**D**) AST, (**E**) glucose, and (**F**) insulin (*n* = 5). * *p* < 0.05, ** *p* < 0.01 and *** *p* < 0.001 Control vs. HFD-fed WT mice of the respective weeks. TG: triglyceride, TC: total cholesterol, ALT: alanine aminotransferase, AST: aspartate aminotransferase, HFD: high-fat diet.

**Figure 3 biomedicines-10-00072-f003:**
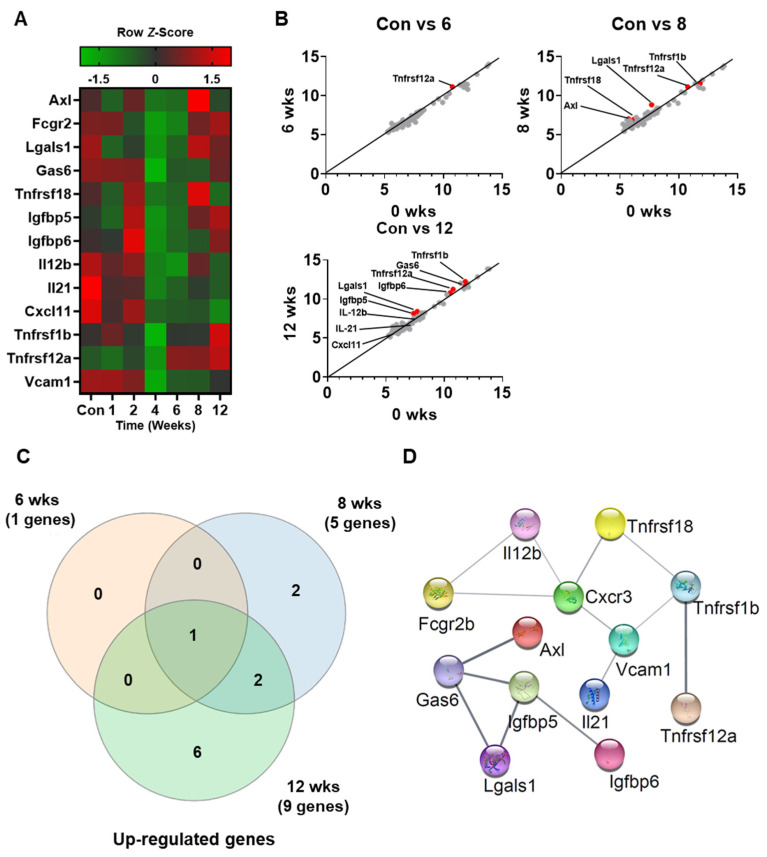
Secretome profiling of differentially expressed cytokines from the serum of control and HFD-fed WT mice: (**A**) Representative cytokine array analysis of the expression of 13 cytokines in the serum from control and HFD-fed WT mice (*n* = 5). (**B**) Scatterplot of the differentially expressed cytokines in control versus 6, 8, and 10 weeks. (**C**) Venn diagram displaying the number of different proteins among the three groups. (**D**) STRING gene networks of interactions of increased cytokines during HFD.

**Figure 4 biomedicines-10-00072-f004:**
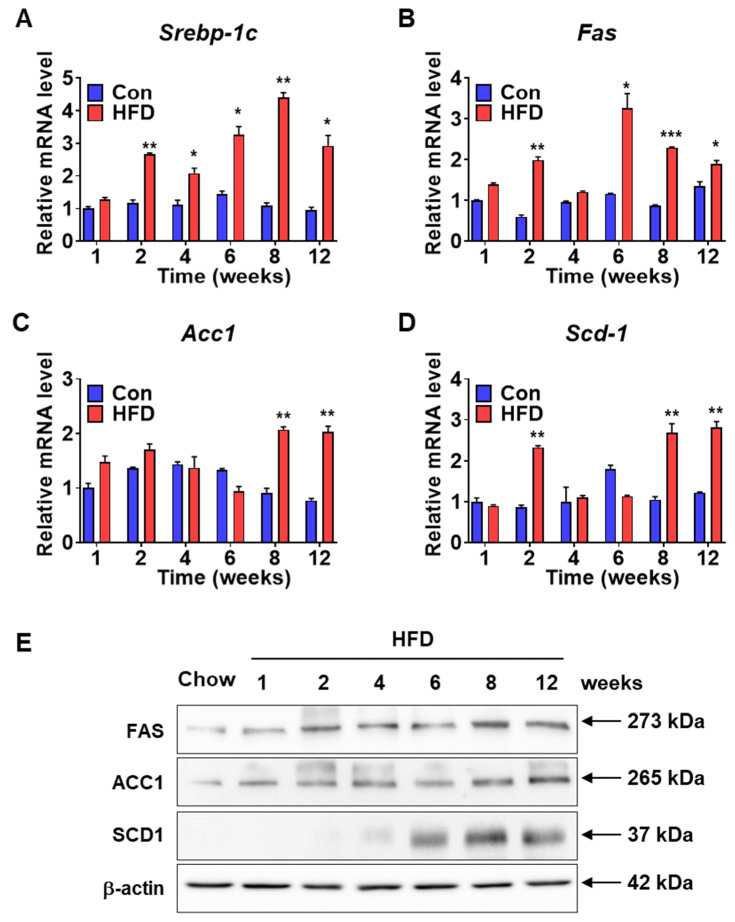
Lipogenic gene expression in the livers of control and HFD-fed WT mice: (**A**–**D**) Expression levels of lipogenesis marker genes (Srebp-1c, Fas, Acc1, and Scd1) in the liver were assessed by qPCR (*n* = 5). (**E**) Western blot data of FAS, ACC1, and SCD1 from the livers of WT mice when on chow diet and HFD (*n* = 5). * *p* < 0.05, ** *p* < 0.01 and *** *p* < 0.001 control vs. HFD-fed WT mice at the respective weeks. Srebp-1c: sterol-regulatory-element-binding protein-1c, Fas: fatty acid synthase, Acc1: acetyl-CoA Carboxylase 1, Scd1: stearoyl-CoA desaturase-1.

**Figure 5 biomedicines-10-00072-f005:**
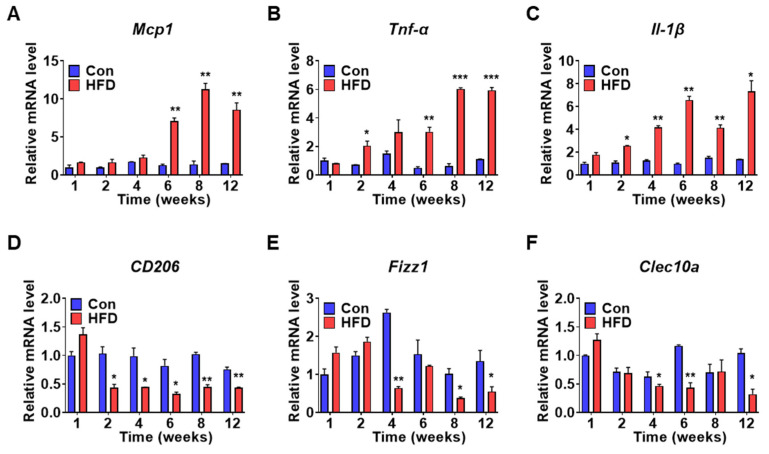
HFD promotes inflammatory gene expression in the liver: (**A**–**C**) mRNA expression levels for pro-inflammatory genes of liver from control and HFD-fed WT mice (*n* = 5). qPCR was used for analyzing Mcp1, Tnf-α, and Il-1β mRNA expression. (**D**–**F**) mRNA levels of anti-inflammatory genes, CD206, Fizz1, and Clec10a in the liver (*n* = 5). * *p* < 0.05, ** *p* < 0.01, and *** *p* < 0.001 control vs. HFD-fed WT mice of the respective weeks. Mcp1: monocyte chemoattractant protein 1, Tnf-α: tumor necrosis factor-alpha, Il-1β: interleukin-1 beta, CD206: mannose receptor c-type 1, Fizz1: inflammatory zone 1, Clec10a: c-type lectin domain containing 10, member A.

**Figure 6 biomedicines-10-00072-f006:**
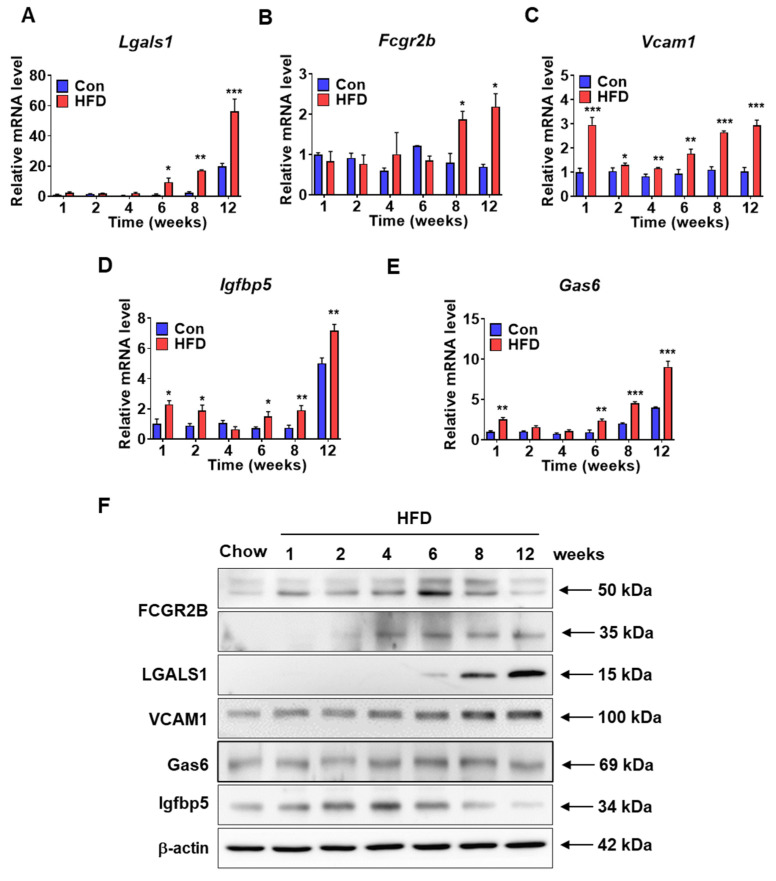
Expression levels of the selected pre-DM bio-markers in the liver samples of control and HFD-fed WT mice: (**A**–**E**) mRNA expression of Fcgr2b, Lgals1, Vcam1, Igfbp5, and Gas6 in the liver from control and HFD-fed WT mice (*n* = 5). (**F**) Protein lysates from the livers of control and HFD-fed WT mice were immunoblotted with indicated antibodies. β-actin was used as a loading control (*n* = 5). * *p* < 0.05, ** *p* < 0.01, and *** *p* < 0.001 control vs. HFD-fed WT mice at the respective weeks. Fcgr2b: Fc receptor, IgG, low affinity Iib, Lgals1: lectin, galactose binding, soluble 1, Vcam1: vascular cell adhesion molecule 1, Igfbp5: insulin-like growth factor binding protein 5, Gas6: growth arrest-specific 6.

**Figure 7 biomedicines-10-00072-f007:**
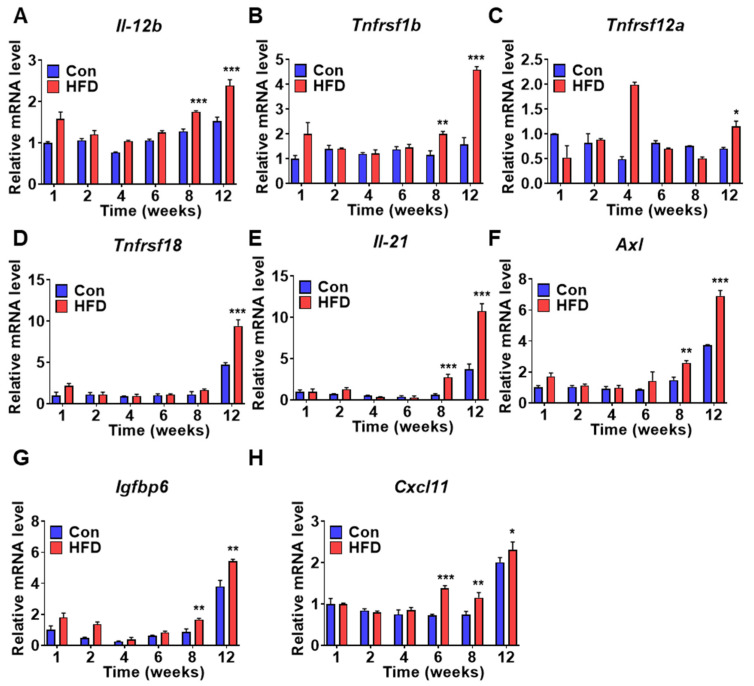
Diabetes marker gene expression is induced in T2DM stage after HFD: (**A**–**H**) Expression of diabetes marker genes was promoted in HFD-fed WT mice. mRNA levels of Il-12b, Tnfrsf1b, Tnfrsf12a, Tnfrsf18, Il-21, Axl, Igfbp6, and Cxcl11 from liver tissue were measured by qPCR analysis (*n* = 5). * *p* < 0.05, ** *p* < 0.01, and *** *p* < 0.001 control vs. HFD-fed WT mice of the respective weeks. Il-12b: interleukin-12b, Tnfrsf1b: TNF receptor superfamily member 1b, Tnfrsf12a: TNF receptor superfamily member 12a, Tnfrsf18: TNF receptor superfamily member 18, Il-21: interleukin-21, Axl: AXL receptor tyrosine kinase, Igfbp6: Insulin-like growth factor binding protein 6, Cxcl11: chemokine (C-X-C motif) ligand 11.

**Table 1 biomedicines-10-00072-t001:** Primer sequences of genes used in qPCR.

Name		Title 3
*L32*	F R	ACATTTGCCCTGAATGTGGT ATCCTCTTGCCCTGATCCTT
*Srebp-1c*	F R	GGAGCCATGGATTGCACATT GGCCCGGGAAGTCACTGT
*Fas*	F R	AAGTTGCCCGAGTCAGAGAA CGTCGAACTTGGAGAGATCC
*Acc1*	F R	TGACAGACTGATCGCAGAGAAAG TGGAGAGCCCCACACACA
*Scd1*	F R	CCGGAGACCCCTTAGATCGA TAGCCTGTAAAAGATTTCTGCAAACC
*Tnf-α*	F R	AGGGTCTGGGCCATAGAACT CCACCACGCTCTTCTGTCTAC
*Mcp1*	F R	ACTGAAGCCAGCTCTCTCTTCC TTCCTTCTTGGGGTCAGCACAG
*Il-1β*	F R	AGCTTCAAATCTCGCAGCAG TCTCCACAGCCACAATGAGT
*Fizz1*	F R	CCAATCCAGCTAACTATCCCTCC ACCCAGTAGCAGTCATCCCA
*Clec10a*	F R	TGAGAAAGGCTTTAAGAACTGGG GACCACCTGTAGTGATGTGGG
*Cd206*	F R	TGTGGTGAGCTGAAAGGTGA CAGGTGTGGGCGCAGGTAGT
*Vcam1*	F R	ACGAGGCTGGAATTAGCAGA TTCGGGCACATTTCCACAAG
*Fcgr2b*	F R	CAAGCAGCAGCCCATACAAT CCCAATGCCAAGGGAGACTA
*Lgals1*	F R	AACCTGGGGAATGTCTCAAAGT GGTGATGCACACCTCTGTGA
*Axl*	F R	GGAACCCAGGGAATATCACAGG AGTTCTAGGATCTGTCCATCTCG
*Cxcl11*	F R	ATCCTGGGAACGTCTGACTG AATACGTGGCTGCATGTTCC
*Il-21*	F R	ATAGGCTCTCGTTCCCACAG CTCCTCAGTCCTCGGGAATC
*Il-12b*	F R	GGGTGACTTAATCGCCACAC CATGGCACGATGGAACTTGT
*Igfbp5*	F R	TTGACCAGCCAGAACAAAGC GACCTCCTCCGTATCCTGTG
*Igfbp6*	F R	CCGCAGACACTTGGATTCAG GACACTGCTGCTTTCGGTAG
*Gas6*	F R	ACAGACTCAGACACCTGTGG GGTCTTCTCCTTGGAGCTGT
*Tnfrsf1b*	F R	CACAGAGGCCCTTCAGGTTA ACAGAGCAACTCTGCCTGAT
*Tnfrsf12a*	F R	GGATTCGGCTTGGTGTTGAT CAGTCCATGCACTTGTCGAG
*Tnfrsf18*	F R	CACTGCCCACTGAGCAATAC CACATGTGTTGCCTCCTCAG

*Acc1*, acetyl-CoA Carboxylase 1; *Axl*, AXL receptor tyrosine kinase; *Cd206*, mannose receptor c-type 1; *Clec10a*, c-type lectin domain containing 10, member A; *Cxcl11*, chemokine (C-X-C motif) ligand 11; *Fas*, fatty acid synthase; *Fcgr2b*, Fc receptor, IgG, low affinity Iib; *Fizz1*, inflammatory zone 1; *Gas6*, growth arrest specific 6; *Igfbp5*, Insulin-like growth factor binding protein 5; *Igfbp6*, Insulin-like growth factor binding protein 6; *Il-1β*, interleukin-1 beta; *Il-12b*, interleukin-12b; *Il-21*, interleukin-21; *Lgals1*, lectin, galactose binding, soluble 1; *L32*, ribosomal protein L32; *Mcp1*, monocyte chemoattractant protein 1; *Scd1*, stearoyl-CoA desaturase-1; *Srebp1-c*, sterol-regulatory-element-binding protein-1c; *Tnf-α*, tumor necrosis factor alpha; *Tnfrsf1b*, TNF receptor superfamily member 1b; *Tnfrsf12a*, TNF Receptor Superfamily Member 12a; *Tnfrsf18*, TNF Receptor Superfamily Member 18; *Vcam1*, Vascular cell adhesion molecule 1.

**Table 2 biomedicines-10-00072-t002:** Gene expression profile in the tissues secretome.

Gene	Full Name	Gene Expression Profiling
*Axl*	AXL receptor tyrosine kinase	adipose white, liver, adipose brown
*Cxcl11*	chemokine (C-X-C motif) ligand 11	adipose white, liver, adipose brown
*Fcgr2b*	Fc receptor, IgG, low affinity Iib	adipose white, liver, adipose brown
*Igfbp5*	Insulin-like growth factor binding protein 5	adipose white, liver, adipose brown
*lgfbp6*	Insulin-like growth factor binding protein 6	adipose white, liver, adipose brown
*lL-12b*	Interleukin 12b	adipose white, liver, adipose brown
*lL-21*	Interleukin 21	adipose white, liver, adipose brown
*Gas6*	growth arrest specific 6	adipose white, liver, adipose brown
*Lgals1*	lectin, galactose binding, soluble 1	adipose white, liver, adipose brown
*Tnfrsf18*	tumor necrosis factor receptor superfamily, member 18	adipose white, liver, adipose brown
*Tnfrsf12a*	tumor necrosis factor receptor superfamily, member 12a	adipose white, liver, adipose brown
*Tnfrsf1b*	tumor necrosis factor receptor superfamily, member 1b	adipose white, liver, adipose brown
*Vcam1*	vascular cell adhesion molecule 1	adipose white, liver, adipose brown

## Data Availability

Please contact to the corresponding author for analyzed data.

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
