# Peer review of "Identification of Pre-Diabetic Biomarkers in the Progression of Diabetes Mellitus"

_biomedicines, 2021, doi:10.3390/biomedicines10010072_

Round 1
Reviewer 1 Report
The authors have responded with satisfaction to this reviewer's comments. Congratulations on the work done.
Author Response
I appreciate the reviewer for accepting our manuscript to publish.
Reviewer 2 Report
The paper “Identification of pre-diabetic biomarkers in the progression of 2 diabetes mellitus" by Lee et al. is a prospective study on mice with the aim of identifying pre-diabetic biomarkers involved in diabetes development and progression.
The article is well written, and only minor spell check is needed. The study has a good design. The article is logically divided into sections and subsections. There are several tables and figures of good quality. The references cited are relevant and adequate. The work has an average degree of novelty and of good interest to the readers. Discussion should be improved as most of data are reported but not adequately commented, also in the light of both pathophysiological and clinical evidence.
Comments:
1) Discussion: increased TG levels are part of the pre-DM and DM disease as it is an indirect sign of insulin resistance, which is also important in NAFLD development and its progression to NASH (DOI: 10.3390/pr9010135). Increased serum transaminases levels are also explained by the low-grade inflammation present in DM, which is one of the main pathophysiological mechanisms in NASH development. The low-grade inflammation also underlined by pro-inflammatory cytokines and genes expression found by the authors is also implicated into the development of endothelial dysfunction, which can also be present in pre-DM patients, thus highlighting the importance of detecting pre-DM as fast as we can, as complications starts to develop in this phase (DOI: 10.3389/fmed.2021.695792).
Author Response
Reply to Reviewer:
The paper “Identification of pre-diabetic biomarkers in the progression of 2 diabetes mellitus" by Lee et al. is a prospective study on mice with the aim of identifying pre-diabetic biomarkers involved in diabetes development and progression.
The article is well written, and only minor spell check is needed. The study has a good design. The article is logically divided into sections and subsections. There are several tables and figures of good quality. The references cited are relevant and adequate. The work has an average degree of novelty and of good interest to the readers. Discussion should be improved as most of data are reported but not adequately commented, also in the light of both pathophysiological and clinical evidence.
Comments:
1) Discussion: increased TG levels are part of the pre-DM and DM disease as it is an indirect sign of insulin resistance, which is also important in NAFLD development and its progression to NASH (DOI: 10.3390/pr9010135). Increased serum transaminases levels are also explained by the low-grade inflammation present in DM, which is one of the main pathophysiological mechanisms in NASH development (DOI: 10.31083/J.RCM2203082). The low-grade inflammation also underlined by pro-inflammatory cytokines and genes expression found by the authors is also implicated into the development of endothelial dysfunction, which can also be present in pre-DM patients, thus highlighting the importance of detecting pre-DM as fast as we can, as complications starts to develop in this phase (DOI: 10.3389/fmed.2021.695792).
Ans: Thank you for your nice comments and kind suggestion. We checked spell in our manuscript again and modified our discussion part based on the reviewer’s comment including references like below.
Increased TG levels indicate pre-DM and DM disease, as it is an indirect sign of insulin resistance, which is also responsible for NAFLD development and its progression to non-alcoholic steatohepatitis (NASH) [24].
Increased serum transaminase levels are also explained by the low-grade inflamma-tion present in DM, which is one of the main pathophysiological mechanisms under-lying NASH development [30]. Low-grade inflammation, caused by pro-inflammatory cytokines, is also implicated in the development of endothelial dysfunction, which can occur in pre-DM patients, thus highlighting the importance of detecting pre-DM as quickly as possible, as complications begin to develop in this phase [31].
- Caturano, A.; Acierno, C.; Nevola, R.; Pafundi, P.C.; Galiero, R.; Rinaldi, L.; Salvatore, T.; Adinolfi, L.E. Sasso, F.C. Non-Alcoholic Fatty Liver Disease: From Pathogenesis to Clinical Impact. Processes 2021, 9, 1, 135, doi:10.3390/pr9010135.
- Galiero, R.; Caturano, A.; Vetrano, E.; Cesaro, A.; Rinaldi, L.; Salvatore, T.; Marfella, R.; Sardu, C.; Moscarella, E.; Gragnano, F., et al. Pathophysiological mechanisms and clinical evidence of relationship between Nonalcoholic fatty liver disease (NAFLD) and cardiovascular disease. Rev Cardiovasc Med 2021 22, 3, 755-768, doi:https:10.31083/j.rcm2203082.
- Salvatore, T.; Pafundi, P.C.; Galiero, R.; Albanese, G.; Di Martino, A.; Caturano, A.; Vetrano, E.; Rinaldi, L.Sasso, F.C. The Diabetic Cardiomyopathy: The Contributing Pathophysiological Mechanisms. Front Med (Lausanne) 2021, 8, 695792. Doi:10.3389/fmed.2021.695792.
And changes made in the manuscript are written using Red.

Round 2
Reviewer 2 Report
I am satisfied with authors replay.
Check line 572, there is a 1. which I think is a mistake.
This manuscript is a resubmission of an earlier submission. The following is a list of the peer review reports and author responses from that submission.
Round 1
Reviewer 1 Report
In the manuscript entitled Identification of pre-diabetic biomarkers in the progression of 2 diabetes mellitus, the authors try to identified novel pre-DM biomarkers which may facilitate the identification of pre-DM patients and help to delay or prevent the progression of T2DM. This is a very good idea since the identification of markers in prediabetes can improve the diagnosis and prognosis of diabetic patients. However, the study is not well designed and therefore does not have the appropriate quality criteria to be published.
Line 53. The authors state that “The progressive stages of T2DM in HFD models are classified as 6 weeks 53 (pre-DM), 12 weeks (T2DM), and 20 weeks (late-stage T2DM) (13)”. In the opinion of this referee, this classification is supported by a single study performed in a rat model fed a high-fat diet. The authors cannot establish this classification based only on one study, they must clarify the model in which it has been established and whether it is extrapolable to other experimental models fed with HFD. In addition, the model used by the authors in the present manuscript are mice fed HFD, not Zucker diabetic fatty (ZDF) rats, thus clarification becomes more important. It cannot be extrapolated that the model used by the authors in this study responds to the above classification.
The authors assume that their model is in a prediabetes state but do not present the metabolic characterization of the animals after 12 weeks of diet. The authors have to present as initial figure measurements of glucose tolerance, glycemia, and insulinemia: always in comparison with a control group fed a standard diet for the same time. Based on this data they should justify why their model is in a prediabetic state, also taking into account that in the introduction they cite a study in rodents in which 12 weeks of HFD results in a diabetic phenotype and not a prediabetic one. Demonstrating that their model is prediabetic is basic to the study performed.
Figure 1. Monitoring of body weight, fat mass, and lean body mass after 12-week high-fat diet. The authors must present data compared to a control group: WT animals on a standard diet for the same time as the HFD group, run in parallel. How do the authors ensure that the mice show a significant weight gain due to the diet? Results cannot be presented without a control group. The comparison following a good scientific method should be made with a control group at the same feeding time, it cannot be assured that the changes in the weight of the animals are due to the diet if this reference group does not exist.
Figure 2. How do the authors know what the normal reference values are at different times if they do not compare with a control group on a standard diet? How can the authors be sure that these changes are due to the diet and not to other factors? It is essential to redesign the study by introducing a WT control group fed with the standard diet.
The same criticism can be extended to the rest of the results. There is no control group with which to compare the results obtained, the only conclusion the authors can draw from the study is that there are biochemical parameters and potential biomarkers that change over time. This does not ensure that they are different from a standard diet, as we do not know this information. Moreover, the authors did not perform any test to ensure that the animals are glucose intolerant by induction and hyperinsulinemic, without affecting fasting or basal glucose (which would show that the animals are in a pre-diabetic state).
This referee cannot recommend the publication of this manuscript in this journal: The model is not well laid out, the control group is missing and measurements to ensure the metabolic stage are missing. The general recommendation is to repeat the entire study with a correct initial design including a group of WT mice fed with a standard diet, a group that should be characterized in parallel to the HFD-fed group.
Reviewer 2 Report
In this manuscript, that authors demonstrated the changes in various biochemical indexes in the mice within 12 weeks by a STZ-HFD animal model. They measured including body weight/fat, serum markers/cytokines, and hepatic genes in their research. They found that during the 12 weeks, some indexes seemed to change with the progression of the disease, so they thought that these indexes might serve as novel T2D biomarkers. Basically, the core pathological cause of T2D is insulin resistance. However, the analysis targets selected by authors do not seem to cover this part, which makes the rationality of its experimental design doubtful. Moreover, using pathological analysis of liver tissue as biomarkers for T2D is not practical for translating experimental results to the human body in the future. Most of the objects analyzed by the authors have also been reported in the previous literature, which also reduces their novelty. I think the author should provide strong evidence to show which factors are indeed the potential biomarkers of pre-DM, and must also consider the feasibility of translating them into the human body. Some of my comments are also attached below for authors’ reference.
- Fig. 1: The authors do not seem to describe how to measure fat mass and lean mass at various points in time? In addition, although statistical differences were marked, the error should still be displayed in the figure.
- Fig. 2F: The result of processing STZ should be the destruction of pancreatic beta cells. However, this figure shows that the insulin content in the blood still increases rapidly and significantly after the 8th week. Does it mean that the authors' STZ did not produce the expected effect? Please elaborate.
- Fig. 3: Among the cytokines selected by the 13 authors, are 0vs6, 0vs 8, and 0vs12 statistically different? It doesn’t seem to be seen from the authors’ results.
- Fig. 4E: The phosphorylation changes of ACC are closely related to the regulation of FAS, and it is recommended that the authors should also provide pACC WB blots.
- Fig. 6 and 7: Actually, many biomarkers authors indicated there are changes, are also considered relevant in some animal models of metabolic diseases (such as NAFLD). Because T2D and NAFLD still belong to different disease entities. Therefore, the authors should provide evidence on how to distinguish whether these biomarkers are only related to the pathophysiological changes of the liver, or are indeed T2D markers?